# Transcriptomic Profiling of Adult-Onset Asthma Related to Damp and Moldy Buildings and Idiopathic Environmental Intolerance

**DOI:** 10.3390/ijms221910679

**Published:** 2021-10-01

**Authors:** Hille Suojalehto, Joseph Ndika, Irmeli Lindström, Liisa Airaksinen, Kirsi Karvala, Paula Kauppi, Antti Lauerma, Sanna Toppila-Salmi, Piia Karisola, Harri Alenius

**Affiliations:** 1Occupational Medicine, Finnish Institute of Occupational Health, 00032 Helsinki, Finland; Hille.Suojalehto@ttl.fi (H.S.); irmeli.lindstrom@ttl.fi (I.L.); Liisa.Airaksinen@ttl.fi (L.A.); kirsi.karvala@varma.fi (K.K.); 2Human Microbiome (HUMI) Research Program, Faculty of Medicine, University of Helsinki, 00014 Helsinki, Finland; joseph.ndika@helsinki.fi (J.N.); Piia.Karisola@helsinki.fi (P.K.); 3Varma, 00098 Helsinki, Finland; 4Skin and Allergy Hospital, Helsinki University Hospital, 00250 Helsinki, Finland; paula.kauppi@hus.fi (P.K.); antti.lauerma@hus.fi (A.L.); sanna.salmi@helsinki.fi (S.T.-S.); 5Institute of Environmental Medicine (IMM), Karolinska Institutet, 171 77 Stockholm, Sweden

**Keywords:** adult-onset asthma, building dampness and molds, endotypes, environmental intolerance, transcriptomics, pathobiological mechanisms

## Abstract

A subset of adult-onset asthma patients attribute their symptoms to damp and moldy buildings. Symptoms of idiopathic environmental intolerance (IEI) may resemble asthma and these two entities overlap. We aimed to evaluate if a distinct clinical subtype of asthma related to damp and moldy buildings can be identified, to unravel its corresponding pathomechanistic gene signatures, and to investigate potential molecular similarities with IEI. Fifty female adult-onset asthma patients were categorized based on exposure to building dampness and molds during disease initiation. IEI patients (n = 17) and healthy subjects (n = 21) were also included yielding 88 study subjects. IEI was scored with the Quick Environmental Exposure and Sensitivity Inventory (QEESI) questionnaire. Inflammation was evaluated by blood cell type profiling and cytokine measurements. Disease mechanisms were investigated via gene set variation analysis of RNA from nasal biopsies and peripheral blood mononuclear cells. Nasal biopsy gene expression and plasma cytokine profiles suggested airway and systemic inflammation in asthma without exposure to dampness (AND). Similar evidence of inflammation was absent in patients with dampness-and-mold-related asthma (AAD). Gene expression signatures revealed a greater degree of similarity between IEI and dampness-related asthma than between IEI patients and asthma not associated to dampness and mold. Blood cell transcriptome of IEI subjects showed strong suppression of immune cell activation, migration, and movement. QEESI scores correlated to blood cell gene expression of all study subjects. Transcriptomic analysis revealed clear pathomechanisms for AND but not AAD patients. Furthermore, we found a distinct molecular pathological profile in nasal and blood immune cells of IEI subjects, including several differentially expressed genes that were also identified in AAD samples, suggesting IEI-type mechanisms.

## 1. Introduction

Asthma is a heterogeneous disease driven by interactions between airway epithelium, the immune system, and environmental exposure. It can be sub-classified into several pheno- and endotypes based on clinical, functional, and inflammatory features [1]. Adult-onset asthma is more common in females, is typically nonallergic, and has less favorable prognosis, when compared with early-onset asthma [2,3]. Contrary to early-onset allergic asthma, there is limited understanding of the biology that underlies the adult-onset asthma phenotype.

Some studies have reported an increase of asthma incidence in adults living or working in damp and moldy buildings [4]. Adult-onset asthma initiated during exposure to building dampness and molds does not demographically differ from the adult-onset asthma phenotype; it is more common in females and predominantly non-allergic [5,6]. These patients use asthma medication excessively, suggesting poor prognosis. The underlying mechanisms are poorly known; however, it seems immunoglobin E (IgE)-mediated sensitization to molds is not an essential factor for asthma onset in these environments [5].

Idiopathic environmental intolerance (IEI) (or multiple chemical sensitivity) patients have recurring, non-specific symptoms in multiple organ systems attributed to environmental factors with no medical and exposure-related explanation [7]. IEI is considered a functional somatic disorder of biopsychosocial nature rather than a toxicological response [8,9]. In Finland and other Northern European countries, these patients often attribute their symptoms to buildings [10]. IEI and adult-onset asthma patient groups are demographically similar, they report similar respiratory symptoms, and there is comorbidity between these conditions [11,12].

We aimed to study if a specific clinically identified subset of patients with adult-onset asthma associated with damp and moldy buildings can be distinguished by assessing transcriptomic profiles and corresponding pathobiological mechanisms in nasal mucosa and peripheral blood mononuclear cells (PBMC). In addition, we evaluated if these patients have similar mechanisms of disease to IEI.

## 2. Results

### 2.1. Demographic and Clinical Data

Figure 1 shows a workflow of the study.

All participants were nonsmoking women. BMI, previous smoking, or atopy did not differ significantly between the groups, and the controls were younger than other groups (*p* = 0.021) (Table 1). Six IEI patients had concomitant asthma. Symptom duration, or the proportion of severe asthma or the ICS dose, did not differ significantly between the asthma and IEI groups. Chronic rhinitis was less common in controls than in other groups (*p* = 0.004). QEESI chemical intolerance (*p* < 0.001) and symptom and life impact scores (*p* < 0.001) were highest in the IEI group and lowest in the control group, with the asthma groups’ scores in between. The IEI group reported most disabling symptoms related to muscles or joints, heart or chest, and ability to think. Total IgE, blood eosinophils, fractional exhaled nitric oxide, FEV1, and FEV1/FVC did not differ significantly between the groups. The asthma and IEI patients had more bronchial hyperresponsiveness than the controls (*p* = 0.029).

### 2.2. Inflammatory Cytokine and Gene Expression Profile of Patients and Controls

At the protein level, proinflammatory cytokines IL-8, IL-12(p70), and IL-17A and regulatory cytokine IL-10 did not differ from the controls in any of the tested groups. However, IL-6 was slightly elevated in the plasma of the asthma not associated with dampness and molds (AND) patients (Appendix A).

Based on the global gene expression profiles, there was considerable overlap between the different asthma subgroups (Appendix A). Nasal biopsy and PBMC transcriptomes separated the controls from the asthma patients. However, the distinction of controls from the asthma subgroups was different for the two tissue types. In the nasal biopsy, all the control samples were entirely separated from all the other patient samples, whereas in the PBMC samples, the controls only separated clearly from the asthma possibly associated with dampness and molds (APD) and IEI patient subgroups. When the individual patient groups were compared with the controls, 409 genes in the nasal biopsies and 266 genes in the PBMCs were identified as significantly different. In the nasal biopsy samples, AND appeared to be the most distinct of the asthma groups with 202 differentially expressed genes (DEGs). Within the IEI category, 101 DEGs were identified in the nasal biopsies and 222 DEGs in PBMCs (Figure 2A). DEGs from all patient/control contrasts in each sample type are provided in Appendix A (PBMC DEGs) and Appendix A (nasal biopsy DEGs).

Principal component analysis (PCA) based on log2-transformed gene expression intensities of identified DEGs from nasal biopsy or PBMC, separated the study samples into three and two main clusters, respectively (Figure 2B,C). When disease and ICS dose are incorporated into a PCA plot constructed from the patient/control DEGs in the nasal biopsies, the clustering of the samples is not explained by differences in ICS dose (Appendix A). The systemic effects of ICS were taken into account during analysis of differentially expressed genes in PBMCs. A subtle separation of the asthma groups from the controls was observed from PCA analysis of DEGs. The APD and AND groups clustered closest to the controls, and just like in the nasal biopsies, asthma associated with dampness and molds (AAD) patients were closest (50% overlap) to the IEI cluster (Figure 2C). This IEI cluster (100% IEI + 5% Ctrl) was clearly distinct from the Ctrl cluster (95% Ctrl) (Figure 2C).

### 2.3. Biological Significance of Differentially Expressed Genes

Venn comparisons of identified DEGs suggested very little overlap (96% of DEGs were unique to each asthma patient subgroup) in disease pathobiological mechanisms in blood cells (Figure 3A). Only AAD and IEI patients had common DEGs—12 in total. On the other hand, the degree of overlap was notably higher in the patient/control DEGs identified from nasal biopsies (Figure 3B). With 27% DEGs in common, the AND and APD asthma subtypes were the most closely related asthma groups in this study. AAD had more DEGs in common with IEI (20%) than either APD (8%) or AND (6%). Nineteen DEGs were commonly shared between all three asthma groups.

In nasal biopsy samples, due to the limited number of shared DEGs between the different asthma subtypes, the unique DEGs for each asthma group were analyzed for enrichment of potentially unique pathobiological mechanisms. Although 51 (AAD), 64 (APD), and 100 (AND) genes were uniquely differentially expressed in the associated patient subgroup, these genes were not significantly associated with any known biological processes or pathways. Therefore, we proceeded with functional enrichment analysis of all the identified DEGs in each asthma subtype. Analysis of the identified DEGs for functional enrichment based on gene ontology (GO) biological processes revealed clear differences between the patient groups and the target tissues. Due to the low number of DEGs, no enriched biological processes were observed in any of the asthma groups in blood cells. In sharp contrast, the IEI group demonstrated very significant (adjusted *p*-value, 1×10^−1^ to 1×10^−7^) enrichment of biological processes such as “*cytokine-mediated signaling pathway*”, “cellular *response to cytokine stimulus*”, “*regulation of IL2 production*”, and “cellular *response to lipopolysaccharide*”, “*inflammatory response*”, etc. The top 20 overrepresented GO biological processes and their corresponding *p*-values are shown in Figure 3C. On the other hand, the IEI group DEGs from the nasal biopsy transcriptome did not show any enrichment of biological processes. In the nasal biopsy samples, significantly enriched (adjusted *p*-value < 0.05) GO biological processes were only identified from DEGs in the AND and APD groups. The top enriched biological processes in the AND group were “*amide transport*”, “response to organophosphorous” and “muscle contraction”. The top 20 GO biological processes overrepresented in the AND subgroup DEGs are shown in Figure 3D, upper panel. Only three biological processes (“*epidermis development*” and “*defense response to bacterium*” and “*amide transport*”) were enriched in the APD group (Figure 3D, lower panel). No significant enrichments of biological processes were observed in the AAD asthma group.

For the patient subgroups demonstrating DEGs with significantly enriched GO biological processes (i.e., IEI for PBMC and APD, AND for nasal biopsies), we next sought to identify specific co-expressed gene networks (modules) for each clinical asthma subtype using the INfORM tool for module prioritization and Ingenuity’s pathway analysis (IPA) for identification of up and downregulated downstream biological functions within each co-expressed gene network. In PBMC, three gene modules from IEI DEGs were identified. Significantly enriched pathways were identified from modules 1 (M1) and 3 (M3), all of which were predicted to be downregulated (Z-score < 0). The most significant biological processes represented by co-expressed genes in M1 were decreased recruitment, migration, and activation of blood cells/leukocytes (B-H adjusted *p*-value < 1×10^−12^), and in M3, decreased proliferation and migration of blood cells/lymphocytes (B-H *p*-value < 1×10^−7^). The interaction between all IEI-associated gene network modules and their corresponding downstream biological processes are depicted in Figure 4.

From the nasal biopsy DEGs, five modules (M1–M5) associated with the AND asthma subtype were identified. All five gene modules consisted of significantly enriched biological processes, some of which were predicted as activated (Z-score > 0) or inhibited (Z-score < 0). The most significant biological processes were identified in modules M2 (decreased *segregation of chromosomes*; B-H *p*-value < 1×10^−13^, *mitosis*; B-H *p*-value < 1×10^−12^) and M4 (increased *smooth muscle contraction*; B-H *p*-value < 1×10^−9^, *formation of filaments*; B-H *p*-value < 1×10^−4^). The interaction between gene network modules associated with the AND asthma subgroup, their corresponding overrepresented biological functions, and predicted activation states are shown in Figure 5. The most significantly enriched function in module M1 was *secretion of triacyl glycerol* (B-H *p*-value < 1×10^−3^), *metabolism of prostaglandin* (B-H *p*-value < 1×10^−3^) in M3, and *quantity of MHC Class I on cell surface* (B-H *p*-value < 1×10^−2^) in module M5.

Four modules were also found to be associated with nasal biopsy DEGs in the APD asthma subtype. Module interaction and their overrepresented biological functions are depicted in Appendix A. Enriched biological functions were identified from three (M2, M3, and M4) of the four co-expressed gene networks. The most significant functions were transport of vitamin/folic acid (M2), interphase/mitosis (M3), and keratinization/differentiation of keratinocytes (M4).

### 2.4. Cell Type Analysis Based on Gene Expression Signature and Flow Cytometry Indicate Macrophages Are Key Mechanistic Players in IEI Pathobiology

Because the functional indications from DEGs in the blood cells of the IEI patients were decreased cell migration, proliferation, and movement, we sought to answer whether these can be substantiated by alternative enrichment analysis. We performed cell type prediction analysis based on DEGs identified in PBMCs of the IEI subjects, as well as flow cytometry cell type profiling. Genes identified as differentially expressed in the IEI subjects relative to the controls were submitted to a publicly available database of RNA-seq data (ARCHS^4^) [14]. In this study, 1716 macrophage samples were identified to be highly enriched (adj. *p* value 4.3×10^−25^) for 40% (88/222 genes) of the IEI/Ctrl DEGs (Figure 6A). Interestingly, elevated monocyte levels in circulation of the IEI patients was the most significant finding from flow cytometry analysis (Figure 6B). A heatmap based on the IEI/Ctrl DEGs identified as signature macrophage genes (88 genes) shows that 94% of these genes were downregulated in the IEI patients (Figure 6C).

### 2.5. QEESI Score and Disease Duration Have the Strongest Correlations to DEGs Identified in Blood Cells

Since the most significantly enriched biological processes were observed from blood cell DEGs, we next sought to investigate whether the DE genes in PBMC are significantly associated to any of the asthma-relevant clinical parameters. To identify genes with specific clinical relevance, Pearson’s correlation analysis was carried out between DEGs identified in blood cells and several clinical parameters in this cohort. A heatmap of genes having a correlation score R > |0.5| to at least one clinical finding is shown in Figure 7A. In total, 49 genes were identified with an R median of −0.52 (QEESI life impact), −0.51 (QEESI chemical intolerance), and −0.33 (symptom duration). These 49 genes (Appendix A) were strongly predicted (Z-score > |2|) to cause decreased immune cell proliferation and migration, decreased cellular homeostasis, decreased cytotoxicity of T-lymphocytes, and decreased activation of antigen-presenting cells (Figure 7B).

The top anti-correlated genes were identified as *NINJ1* (R = −0.59, *p <* 0.0001) and *SQLE* (R = −0.62, *p <* 0.0001). NINJ1 (Ninjurin1) and SQLE1 (squalene epoxidase) were significantly anti-correlated to QEESI life impact and chemical intolerance findings (Appendix A). QEESI life impact and chemical intolerance scores were lowest in the controls and expression of NINJ1 and SQLE were highest in the controls (Appendix A).

## 3. Discussion

There is a lack of knowledge on the mechanisms of adult-onset asthma and IEI [1,15,16]. We used a combination of clinical assessment and global transcriptomic analysis of blood and airway epithelium to investigate whether specific pathobiological mechanisms of adult-onset asthma associated with dampness and molds could be identified. The asthma patients were divided into three groups based on exposure and symptoms to building dampness and molds during disease initiation. Furthermore, we included a group of IEI patients to investigate potential molecular similarities with asthma patients.

The study population represented nonsmoking women, and the clinical characteristics of asthma patients were similar to female and obesity-related asthma phenotypes identified earlier in the cluster analysis of adult-onset asthma [17,18]. Asthma was mainly mild or moderate [13]. Previous tests had confirmed asthma diagnosis of asthma patients and regular ICS use could explain near-normal lung function results during sampling. Atopy and markers of eosinophilic inflammation (blood eosinophils, fractional exhaled nitric oxide, total IgE) did not differ between asthma patients and healthy controls, suggesting non-T2 type disease [1]. Because IEI and asthma commonly overlap [12], it was not possible to completely exclude subjects with an asthma diagnosis from the IEI group. Nevertheless, asthma did not explain the symptoms in that group and QEESI scales showed a clear difference between asthma and IEI groups. To assess IEI, we used chemical intolerances and life impact scales of the QEESI questionnaire, which have shown the largest discriminatory validity of QEESI scales [19]. The IEI patients had experienced symptoms for several years and reported high QEESI scores suggesting chronic and severe IEI [20]. A previous study showed that asthma does not substantially change QEESI results [19]. Higher QEESI scales of asthma groups than the control group suggest more IEI-type symptomatology among asthma patients. This is in line with previous findings showing an overlap between these two entities [12].

We selected nasal epithelia as a local sampling site because in previous studies the quality of nasal epithelia samples has been sufficient for transcriptomic analysis [21], whereas quality of sputum samples varies [22]. Previous studies have identified IL-17, IL-8, and IL-6 as key cytokines in non-T2 asthma [23]. Increased levels of IL-6 have also been found in serum [24] of asthmatic patients and in BALF from patients with nonallergic asthma compared with that from patients with allergic asthma [25]. In the present study, none of these cytokines were significantly different on an mRNA level, and on the protein level, only IL-6 was found to be mildly elevated in the AND subgroup suggesting a weak systemic inflammation. Previous studies have failed to show consistent findings related to immunological parameters [26]. Luca et al. reported that IFN-gamma, IL-8, IL-10, MCP-1, PDGFbb, and VEGF were significantly increased in the plasma compared to healthy controls [27]. Dantoft et al. found that plasma levels of IL-1-beta, IL-2, IL-4, and IL-6 were significantly increased in IEI, whereas IL-13 was downregulated [28]. We did not find significant differences in any of the cytokines studied between IEI and the controls. It is therefore possible that larger sample sizes than that used in the present study are needed to reach statistical significance in the plasma cytokine analysis in IEI.

Gene expression profiling of whole blood of the U-BIOPRED cohort, identified 1693 DEGs (referred as severe asthma disease signature—SADS) between patients with severe asthma and healthy controls, whereas the number of DEGs between mild/moderate asthma and healthy controls were fewer [29]. These results demonstrate that the disease severity has major impact on the blood transcriptome and the number of DEGs. We found a relatively low number of DEGs in the blood cells of the adult-onset asthma patient groups, suggesting weak systemic evidence of disease. Compared to PBMC, six times more DEGs were identified from nasal biopsies of our asthma patients suggesting that disease pathology is primarily localized to airway epithelia. Indeed, several biological processes with previously published relevance to asthma, such as contraction of (airway) smooth muscle [30,31] and altered cell division [32,33,34], were identified as highly enriched functions in the AND group. In addition, enrichment of genes involved in leukocyte migration and cytokine production are the major drivers of the immune dysregulation and clinical manifestations of asthma. Given that clinical and transcriptomics findings in the AND group are well in line with published pathomechanisms associated with asthma, we propose that the AND represents the phenotype of non-IgE-mediated adult-onset asthma. The APD group was the closest to the AND group, in terms of shared DEG and affected biological functions, wherein we observed altered mitosis as one of the most significantly affected airway epithelial functions. In contrast, there was no molecular evidence of localized or systemic inflammation in the AAD group. We cannot exclude the possibility that the use of ICS, which may attenuate inflammation, could explain partially our findings of modest to no systemic or localized inflammation in the airway epithelium.

Although the AAD group did not show evidence of active airway inflammation, based on negative findings in GO term enrichment analysis, it revealed modest clustering together with the IEI patients in PCA. This PCA clustering was due to that 33% (12/38) of the DEGs in AAD were shared with IEI in the blood cell transcriptome and 20% (22/109) of the DEGs in AAD were shared with IEI in the nasal biopsies. Thus, although we did not detect a specific molecular endotype for AAD, partial transcriptomic clustering together with IEI may indicate IEI-type disease mechanisms in this group. In our previous study, we did not find significant differences in nasal mucosa or in the blood transcriptome in subjects with respiratory symptoms associated with moisture damage when compared to non-exposed or non-symptomatic persons [35].

Unlike asthma groups, the IEI patients showed strong enrichment of biological processes related to cytokine stimulus or cytokine signaling in GO term analysis of the blood transcriptome. In pathway analysis, all enriched biological functions related to the immune system were markedly suppressed in the IEI patients. Our findings emphasized the possible role for macrophages in IEI pathology since the macrophage signature genes were suppressed in the blood cells. Our results also show that no active inflammation can be detected in the airway epithelia of the IEI patients. Due to the absence of internationally accepted biological or physiological criteria for diagnosing IEI patients, it has been difficult to reconcile pathological mechanisms underlying IEI. Moreover, the physiological mechanisms of functional somatic disorders including IEI are under debate. In several models, stress-related physiology has been considered a key element in symptom perception [36]. Stress modulates immune function depending on the duration of stress [37,38]. Immunomodulatory mechanisms related to chronic stress could be involved in detected immunosuppression of our IEI patients.

The QEESI life impact and chemical intolerance scores across all patients in this cohort were most correlated to a subset of 49 DEGs identified in blood cells. Enrichment of cellular functions with this gene set is consistent with suppressed immune cell activation and underscores the biological relevance of QEESI-linked genes. The top two genes that were significantly correlated to the subject’s QEESI scores are NINJ1 and SQLE. NINJ1 (Ninjurin1; synonym—nerve injury-induced protein) is involved in macrophage activation and migration [39,40,41]. Squalene epoxidase (SQLE) is an essential enzyme for cholesterol biosynthesis, and a clinical target for treatment of hypercholesteremia, cancer, and fungal infections [42]. In future studies, these genes could be used for more specific phenotyping in larger IEI cohorts, in order to extensively investigate disease mechanisms and identify potential therapeutic targets.

As a conclusion, gene signatures for adult-onset asthma without exposure to dampness revealed clear evidence of inflammation in the airways. In contrast, a clear endotype of asthma related to damp and moldy buildings could not be distinguished from existing biological pathways and co-expressed gene networks. Interestingly, blood cell transcriptomics of IEI subjects showed heavy suppression of immune cell functions. Twenty to thirty percent of the DEGs identified in IEI nasal and blood samples were also detected in samples from asthma patients exposed to dampness and molds, suggesting IEI-type mechanisms.

## 4. Materials and Methods

The study methods are briefly described below. Where applicable, additional details are provided in the manuscripts’ online Appendix A.

### 4.1. Exposure Assessment

Participants’ exposure to indoor dampness and mold during the onset of asthma symptoms was evaluated independently by two experts (KK, HS). The assessment was based on indoor air and building evaluation reports, medical records, and workplace reports. The person’s own perception of exposure was also considered. Exposure was classified in three categories: (1) *Substantial exposure to dampness and molds* was found if at home or at the workplace a significant water damage and related microbial growth in the building materials was verified. In this category, a reliable description of a wide-ranging water-damaged structure (with a size of more than 1 m^2^) was available. Fungal spores were found in material samples or in indoor air samples exceeding national limit values [43]. (2) *Possible exposure to dampness and molds* was found when evidence of damage was sparse or unsure. Subjects were categorized in this group also if they reported dampness or mold damage at home or at the workplace during asthma onset, but there was no evidence of the damage. (3) *No exposure to dampness and molds* group had no evidence of exposure to dampness or mold growth at home or at the workplace and the subjects did not report suspicions of these damages.

### 4.2. Study Population and Groups

All subjects were nonsmoking women aged 18–65 years. Asthma and IEI patients were recruited from tertiary hospital asthma and occupational disease outpatient clinics at Helsinki University Hospital and the Finnish Institute of Occupational Health during 2015–2018, and the control group was recruited using advertisements and social media. A medical doctor (HS) went through medical files to verify that the asthma diagnosis was confirmed according to standard procedure, and at least one of the diagnostic criteria were fulfilled: (1) marked ≥12% and 200 mL postbronchodilator increase in FEV_1_ in spirometry, (2) recurrent significant reversibility in diagnostic peak flow monitoring, or (3) non-specific bronchial hyperresponsiveness in histamine or methacholine provocation test. Age of asthma onset, and allergic and asthma symptoms were evaluated from medical files and patients’ interviews. In addition to gender and age, the inclusion criteria of asthma groups were (1) adult-onset asthma (onset of asthma symptoms ≥18 years of age), and (2) no allergic or asthma symptoms related to common environmental allergens. The subjects were classified in the following groups:*Asthma associated with dampness and molds* (*AAD*) had ‘*substantial exposure to dampness and molds*’—during the asthma onset their symptoms deteriorated at the damaged building.*Asthma possibly associated with dampness and molds* (*APD*) had ‘*possible exposure to dampness and molds*’—during the asthma onset their symptoms deteriorated at the workplace or at home.*Asthma not associated with dampness and molds* (*AND*) had ‘*no exposure to dampness and molds*’ or deterioration of asthma symptoms at home or at the workplace during asthma onset.*Idiopathic environmental intolerance* (*IEI*) group inclusion criteria were modified from WHO’s IEI criteria [16] and that of Lacour et al. [44]: (1) respiratory symptoms, (2) symptoms associated environmental factors in low exposure levels which do not cause symptoms to the majority of people, (3) symptoms from several organ systems, (4) recurrent symptoms related to certain environments that diminish or end when the factor related to symptoms is removed, (5) the condition had lasted at least 6 months and causes major living restrictions in ability of function, (6) symptoms cannot be explained by any known somatic or psychiatric disease, (7) the person experienced that the symptoms are caused by environmental factors. Asthma patients were not excluded.*Healthy controls* (*Ctrl*) had no allergic symptoms related to common environmental allergens, asthma, or chronic cough or dyspnea. They had no known long-term working or living in a building with dampness or mold damage. They did not fulfill the criteria of IEI.

Methods of clinical assessments at the time of sampling are presented in the online Appendix A. Chemical intolerances and life impact scales of the Quick Environmental Exposure and Sensitivity Inventory (QEESI) were used to gauge IEI [45].

The study was approved by Helsinki University Hospital Coordinating Ethical Committee (80/13/03/00/15). Each participant gave written informed consent.

### 4.3. Inhaled Steroid Course for Healthy Controls

For safety reasons, inhaled corticosteroids (ICS) were continued during sample collection, and nasal steroids were ceased one month earlier when possible. To control the systemic ICS effect in the analysis, we assessed the ICS effect in the healthy controls. They used high dose ICS (budesonide DPI 800 ug/day) for 5–6 weeks and a PBMC sample was taken before and after the treatment.

### 4.4. Sampling

Nasal biopsies taken from mucosa of inferior concha were stored in RNAlater solution at −80 °C. PBMCs were isolated from venous blood (8 mL) and collected into CPT tubes (BD).

### 4.5. PBMC Separation and Flow Cytometry

Separated plasma was aliquoted and stored at −20 °C. The extracted PBMCs were frozen in cell freezing medium (Gibco) and stored in a deep freezer (−80 °C). The number of total cells was counted, and the relative proportions of cell populations was determined by flow cytometry using surface markers for T cells, B cells, NK cells, and monocytes.

### 4.6. Cytokine Profiling by Luminex Assay

The soluble inflammatory markers including IL-1ra, IL-6, IL-8, IL-10, IL-12(p70), and IL-17A were measured from plasma by Bio-Plex Pro Assays (Bio-Rad Corporation, Hercules, CA, USA) in a Luminex (Bio-Plex 200, Bio-Rad) system according to the provided instructions.

### 4.7. Transcriptomics

Total RNA was isolated from biopsy samples on the RNeasy Plus Mini Kit (QIAGEN, Hilden, Germany), and RNA from the white blood cells was isolated by RNA AllPrep DNA/RNA/miRNA Universal Kit (QIAGEN). The amount and quality of RNA was checked and confirmed. Briefly, 100 ng of total RNA was amplified, labelled with Cy3 and Cy5 dyes, and hybridized to human microarrays (SurePrint G3, Agilent Technologies, Canta Clara, CA, USA). Raw data were monitored for quality, and quantile normalized (Bioconductor package Limma).

### 4.8. Data Analysis

We used SPSS version 25.0 (IBM Corporation, Armonk, NY, USA) for analysis of demographic and clinical parameters. The differences between the groups were analyzed using the Kruskal–Wallis test for continuous variables and chi-squared tests for categorical variables. The Kruskal–Wallis test was used also for Luminex cytokine analysis followed by Dunn’s test for multiple comparisons when applicable.

For transcriptomics, preprocessing and differential gene analysis were analyzed with *eUTOPIA* bioinformatics packages [46]. Differential gene expression analysis between different test groups was performed by *Limma Model* analysis [47]. Age (biopsy and PBMC), body mass index (BMI) (biopsy and PBMC), and ICS use (PBMC only) were used as co-factors. The multivariate correction of false discovery rate was performed by the *Benjamini–Hochberg* method. A minimum log2 difference of 0.58, together with a maximum adjusted *p* value of 0.05 was implemented as a cut-off to consider a gene as significantly differentially expressed. Based on our experience in toxicogenomics and clinical transcriptomics/proteomics, a combination of 1.5-fold change threshold, FDR correction, and identification of overrepresented biological pathways is a reliable and validated approach to identify biologically relevant differentially expressed genes. *Perseus* and *Chipster* were used to generate gene clusters and heatmaps [48,49]. The top 2 principal components were used to depict clustering of the samples in PCA plots. Heatmap clustering parameters used were as follows; distance: Euclidean, linkage: average and cluster preprocessing: k-means, with the number of clusters set to exceed the total number of samples—default was 300. Gene-phenotype correlations were studied with a Pearson’s correlation. Analysis details are given in Appendix A. All raw data associated with this study, as well as the normalized data matrix, have been submitted into the GEO repository with the accession number: GSE182797, GSE182798. 

### 4.9. Pathway Enrichment Analysis

A three-step process was used to infer the biological relevance of differentially expressed genes. First, the list of differentially expressed genes (DEG) from each patient/control contrast was submitted into Enrichr for identification of overrepresented gene ontology biological processes (GO Biological Process 2021). Fisher’s exact test with a maximum FDR threshold of 5% (adjusted *p* < 0.05) was implemented to consider a biological process as significantly enriched. Next, each set of patient/control DEGs consisting of significantly enriched GO biological processes was submitted to INfORM (Inference of NetwOrk Response Modules) [50]—an R-shiny application wherein gene level expression, fold changes, and differential *p*-values are used to detect, evaluate, and select gene modules with high statistical and biological significance. Finally, the activated or inhibited biological functions represented by each module gene set was determined using a z-scoring algorithm and Fischer’s exact test (FET) implemented in the IPA pathway analysis tool (Ingenuity IPA version 65367011, QIAGEN). In IPA, significant overlaps between INfORM-identified response modules and biological processes (functions) relative to the whole Ingenuity knowledgebase were determined (adjusted *p*-value threshold of 0.05). Z-scoring assessment using the corresponding patient/control expression fold-change of each module gene was then used to predict whether the identified significantly enriched functions were activated or inhibited. Where applicable, we referred to a significantly enriched downstream biological process as activated if the z-score > 0 or inhibited if the z-score < 0. Z-score = 0 implies the direction of the effect is unknown or ambiguous. Z-scores that exceed ±2 are ‘highly predictive’ of an activated or inhibited biological process.

## Figures and Tables

**Figure 1 ijms-22-10679-f001:**
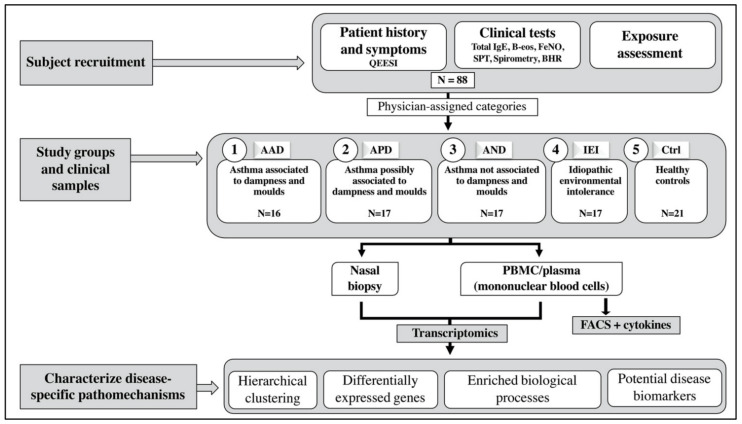
Flowchart outlining sample categorization and study workflow. Clinical tests included spirometry, assessment of non-specific bronchial hyperresponsiveness (BHR), fractional exhaled nitric oxide (FeNO), and blood eosinophil (B-eos) counts. Allergy to common environmental allergens was determined via skin prick testing (SPT). The Quick Environmental Exposure and Sensitivity Inventory questionnaire (QEESI) was used to assess idiopathic environmental intolerance. Nasal biopsies and peripheral blood mononuclear cell (PBMC) samples were obtained. Disease mechanisms were investigated with gene expression profiling, cytokine analysis (ELISA), and immune cell subtype profiling (FACS).

**Figure 2 ijms-22-10679-f002:**
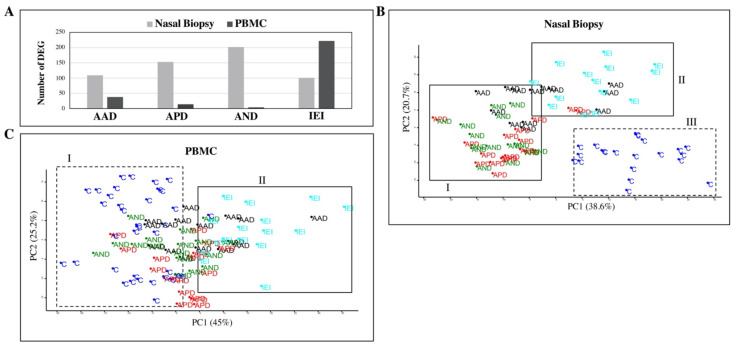
Comparison of differentially expressed genes (DEGs) identified in patient versus controls (**A**). Patient groups with asthma that is *associated* to dampness and molds, *possibly associated* to dampness and molds, and *not associated* with dampness and molds are denoted as AAD, APD, and AND, respectively. Patients diagnosed with idiopathic environmental intolerance are denoted as IEI, and controls are denoted as (**C**). Principal component analysis (PCA) shows top two components represented by genes identified as differentially expressed in the nasal biopsies (**B**) and blood cells (PBMCs) (**C**) of patients versus controls. In the nasal biopsies (**B**), the top 2 components of the PCA plot separated all samples into three main clusters. Cluster I consisted of all AND patients, 90% of all APD patients, and 50% of all AAD patients. Cluster II consisted of all IEI patients and a couple of APD and half of all AAD patients. All control samples grouped together in Cluster III. In blood cells (**C**), clearly distinct asthma patient or control subgroup clusters cannot be identified from the top 2 components of the PCA plot. However, the control samples (Cluster I) do cluster separately from IEI individuals (Cluster II).

**Figure 3 ijms-22-10679-f003:**
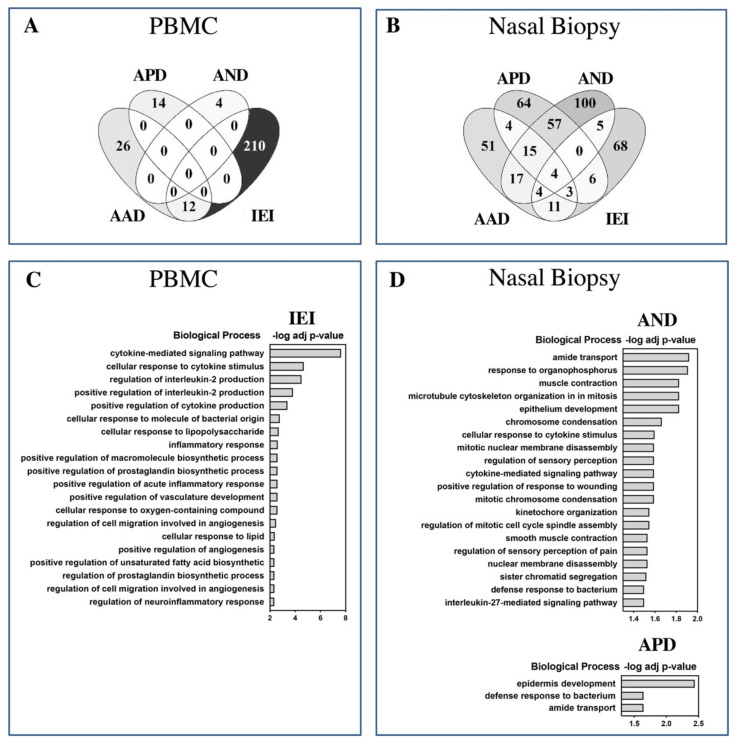
Venn diagram of differentially expressed genes (DEGs) identified in patient versus controls in PBMC (**A**) and nasal biopsy (**B**). Patient groups with asthma that is *associated* to dampness and molds, *possibly associated* to dampness and molds, and *not associated* with dampness and molds are denoted as AAD, APD, and AND, respectively. Patients diagnosed with idiopathic environmental intolerance are denoted as IEI and controls are denoted as ***C***. *AND* and *APD* had the greatest number of shared DEGs in nasal biopsy, while AAD and IEI have the most DEGs in common in blood cells (PBMC). Gene ontology-based analysis of enriched biological processes in different patient groups reveals several significantly overrepresented pathways. The top pathways, ranked by −log adjusted *p* value (cutoff, 1.3) are shown for each differentially expressed gene set identified in PBMC (**C**) and nasal biopsy (**D**).

**Figure 4 ijms-22-10679-f004:**
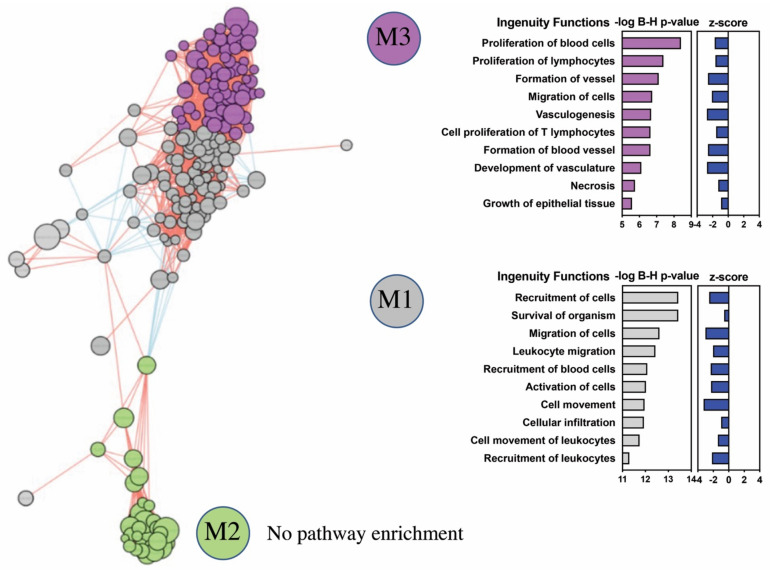
Prediction of the disease relevant pathways represented by differentially expressed genes (DEGs) in peripheral mononuclear cells of patients with idiopathic environmental intolerance (IEI). Three co-expressed gene networks (modules; M1, M2, and M3) were identified from genes that are differentially expressed between IEI patients and controls. The interactions between different inferred co-expressed gene network modules, as well as their corresponding top enriched biological functions and predicted downstream activation states are shown for modules M1 and M3. Biological functions are ranked by −log of Benjamini–Hochberg corrected p values (implemented filter: exclude cancer pathways; include only pathways with ≥ 5 differentially expressed genes). Positive (+) or negative (-) activation scores correspond to downstream activated or inhibited disease/functions, respectively. An activation Z score > |2| is highly predictive of an activated or inhibited disease/function. No enriched biological functions were identified from the genes in module M2.

**Figure 5 ijms-22-10679-f005:**
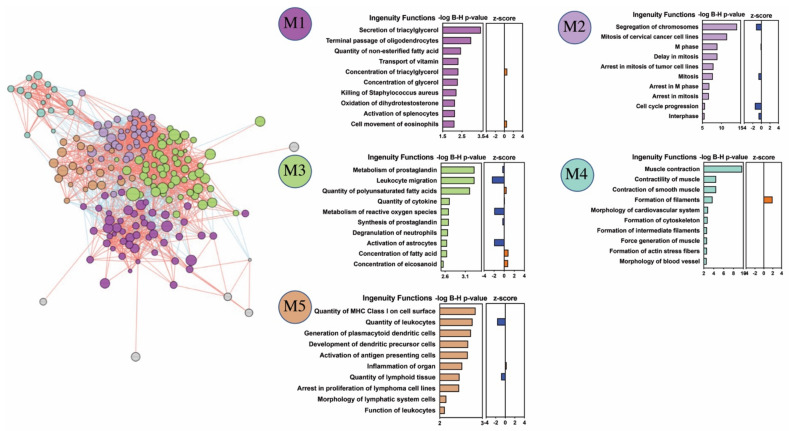
Prediction of the disease relevant pathways represented by differentially expressed genes (DEGs) in nasal biopsy of patients with asthma that is not associated to damp and moldy buildings (*AND*). Five co-expressed gene networks (modules; M1, M2, M3, M4, and M5) were identified from genes that are differentially expressed between *AND* patients and controls. The interactions between different inferred co-expressed gene network modules, as well as their corresponding biological processes and predicted downstream activation states are shown. Biological functions are ranked by −log of Benjamini–Hochberg corrected p values (implemented filter: exclude cancer pathways; include only pathways with ≥5 differentially expressed genes). Positive (+) or negative (-) activation scores correspond to downstream activated or inhibited disease/functions, respectively. An activation Z score > |2| is highly predictive of an activated or inhibited disease/function.

**Figure 6 ijms-22-10679-f006:**
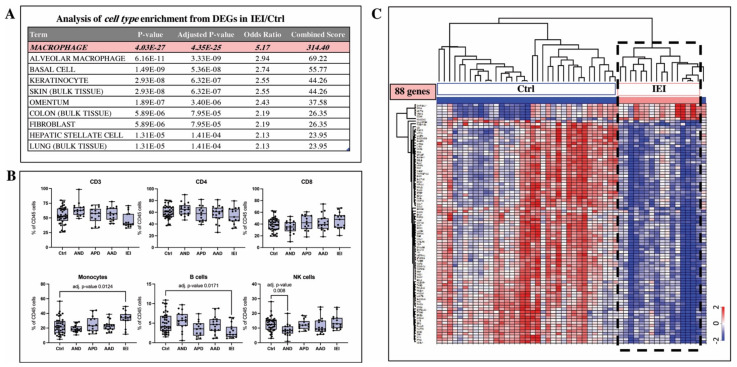
Identified differentially expressed genes (DEGs) in blood cells of idiopathic environmental intolerance (*IEI*) patients relative to controls (Ctrl) were submitted to cell type enrichment analysis. The most significant cell type represented by the expression signature of these genes were macrophages (**A**). The relative expression of 88 genes identified to be differentially expressed between IEI patients and controls was consistent with genes enriched in RNA-seq expression profiles (1716 samples) of macrophages in ARCHS database (overlapping not shown). Cell type enrichment analysis was also performed via FACS analysis. Significantly elevated levels of monocytes were observed in IEI subjects’ blood. Patient groups with asthma that is *associated* to dampness and molds, *possibly associated* to dampness and molds, and *not associated* with dampness and molds are denoted as AAD (n = 16), APD (n = 17), and AND (n = 17), respectively. Patients diagnosed with idiopathic environmental intolerance are denoted as IEI (n = 16) and controls are denoted as ***Ctrl*** (n = 21). Analysis was performed using the Kruskal–Wallis test with Dunn’s post-hoc correction (**B**). Despite having elevated monocyte levels, macrophage signature genes were found to be predominantly downregulated in the blood cells of individuals diagnosed with IEI. A heatmap showing the relative expression (z-score normalized) of these genes in control and IEI blood cell samples is shown in (**C**). Controls comprise individuals who did not take any inhaled corticosteroid (n = 21) plus a subset of controls that used inhaled corticosteroid for 4 weeks (n = 14). The dashed bold rectangle highlights a cluster of IEI patients with downregulated expression of these macrophage-associated genes.

**Figure 7 ijms-22-10679-f007:**
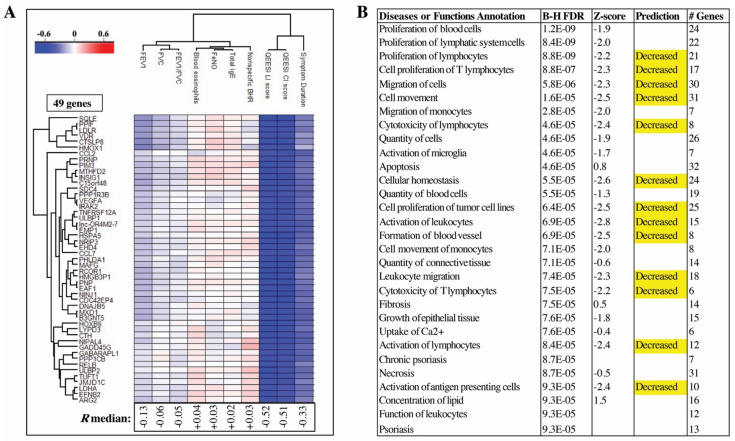
Regression analyses of differentially expressed genes (DEGs) and selected clinical parameters relevant to diagnosis of asthma and/or idiopathic environmental intolerance identified 49 genes with a Pearson’s correlation coefficient of R >|0.5| to at least 1 clinical parameter. A hierarchical cluster of correlated genes (cutoff R > |0.5|) and clinical parameters is shown in (**A**). The correlation coefficients of these 49 genes clusters the clinical parameters into 2 main clusters. The strongest correlations were identified in both the QEESI chemical intolerance (CI) and life impact (LI) scores, wherein expression of all 49 genes was negatively correlated with QEESI scores across the entire study cohort. Pathway analysis of these 49 genes predicted highly significant enrichment of genes involved in suppression of immune cell activation, proliferation, and/or migration (**B**). BHR is non-specific bronchial hyperresponsiveness, CI is chemical intolerance, FeNO is fractional exhaled nitric oxide, FEV1 is forced expiratory volume in one second, FVC is forced vital capacity, IgE is immunoglobin E, LI is life impact, and QEESI is Quick Environmental Exposure and Sensitivity Inventory.

**Table 1 ijms-22-10679-t001:** Demographic and clinical characteristics during the sample collection of the study subjects. All subjects were nonsmoking women.

	Asthma Associated with Dampness and Molds (AAD) n = 16	Asthma Possibly Associated with Dampness and Molds (APD) n = 17	Asthma Not Associated with Dampness and Molds (AND) n = 17	Idiopathic Environmental Intolerance (IEI) ^a^ n = 17	Healthy Controls (Ctrl) n = 21	*p*-Value
Age, years, md (IQR)	47.3 (40.9–53.4)	50.5 (43.7–55.7)	51.8 (38.5–57.4)	50.3 (44.4–55.1)	41.2 (32.8–47.0)	0.021
Body mass index, kg/m^2^, md (IQR)	24.9 (24.1–30.4)	27.3 (24.0–32.9)	29.0 (26.0–34.5)	25.1 (23.7–27.3)	25.5 (21.1–28.7)	0.060
Ex-smoker, n (%)	3 (19)	6 (35)	4 (24)	3 (18)	1 (10)	0.399
Atopy ^b^, n (%)	3 (19)	4 (24)	1 (6)	6 (35)	7 (33)	0.238
Duration of respiratory symptoms, years, md (IQR)	3.0 (2.0–5.0)	5.0 (1.5–12.5)	8.0 (2.0–16.5)	5.0 (2.5–10.5)	NA	0.122
Use of inhaled steroid within one month, n (%)	16 (100)	16 (94)	17 (100)	10 (59)	0 (0)	<0.001
Inhaled steroid dose, n (%) ^c^						0.786
Low	5 (31)	2 (13)	2 (12)	2 (20)	NA	
Medium	6 (37)	6 (38)	7 (41)	3 (30)	NA	
High	5 (31)	8 (50)	8 (47)	5 (50)	NA	
Daily use of short-acting β2-agonist, n (%)	0 (0)	1 (6)	2 (12)	1 (17)	NA	0.696
Severe asthma ^d^, n (%)	3 (19)	4 (24)	5 (29)	1 (17)	NA	0.875
Chronic rhinitis, n (%)	11 (69)	11 (65)	10 (59)	10 (59)	3 (14)	0.004
QEESI chemical intolerance score, md (IQR), n = 87	39 (24–53)	34 (19–55)	29 (9–71)	76 (39–85)	5 (3–15)	<0.001
QEESI symptom score, md (IQR)	42 (16–61)	35 (15–44)	29 (17–47)	51 (32–78)	6 (3–13)	<0.001
QEESI life impact score, md (IQR)	23 (4–52)	22 (9–40)	15 (15–38)	64 (43–88)	2 (0–7)	<0.001
Total IgE kU/L, md (IQR)	27 (15–78)	14 (7–41)	26 (13–69)	20 (12–52)	22 (16–35)	0.690
Blood eosinophils /µL, md (IQR)	175 (93–215)	130 (95–220)	180 (120–250)	120 (120–175)	160 (110–210)	0.918
Fractional exhaled nitric oxide, md (IQR)	13.5 (11.5–22.1)	17.5 (12.3–23.5)	14.5 (9.0–30.5)	11.5 (9.1–20.6)	12.0 (9.3–16.3)	0.445
FEV_1_ % pred, md (IQR)	91 (86–97)	84 (78–100)	85 (70–95)	95 (88–103)	89 (83–98)	0.093
FVC % pred, md (IQR)	93 (86–101)	92 (79–102)	86 (78–81)	96 (94–103)	95 (86–105)	0.033
FEV_1_/FVC, md (IQR)	78 (75–84)	78 (73–83)	80 (75–81)	81 (76–84)	79 (76–82)	0.671
Nonspecific bronchial hyperresponsiveness, n (%), n = 84						0.029
Moderate	4 (29)	3 (18)	1 (7)	1 (6)	0 (0)	
Mild	2 (14)	5 (29)	3 (20)	4 (24)	0 (0)	
No	8 (57)	9 (53)	11 (73)	12 (71)	21 (100)	

FEV_1_ = forced expiratory volume in one second; FVC = forced vital capacity; IQR = interquartile range; IgE = immunoglobin E; GINA = Global Initiative for Asthma; md = median; NA = not available; SPT = skin prick test; QEESI = the Quick Environmental Exposure and Sensitivity Inventory; ^a^ six subjects had asthma diagnosis; ^b^ positive skin prick test to at least one environmental allergen; ^c^ proportion of cases using inhaled steroid within one month; ^d^ ERS/ATS criteria [13].

## Data Availability

The transcriptomics datasets associated with the current study have been deposited in the GEO Omnibus database and are publicly available with accession numbers GSE182797 and GSE182798.

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
