# Peer review of "Transcriptomic Profiling of Adult-Onset Asthma Related to Damp and Moldy Buildings and Idiopathic Environmental Intolerance"

_ijms, 2021, doi:10.3390/ijms221910679_

Round 1

Reviewer 1 Report

The authors investigated transcriptomics profiles of asthma phenotypes distinguished by exposure to damp/mold in indoor environment and IEI, to provide insights on the disease mechanism of asthma and IEI. I have only few comments as follows.

  1. Regarding the study population and endpoint definition
    • No description is provided how the study participants were recruited, especially the control group.
    • It appears that the asthma phenotypes being investigated in this study are adult-onset, non-allergic asthma, which then was categorized by exposure to the indoor damp/mold. However, it is not clear how the “adult-onset” “non-allergic” asthma was defined. Which age criteria and based on what information – based on the self-report or clinical history? Allergy defined by self-report?
  2. PBMC samples were collected from the healthy controls before and after the ICS treatment. It is not clear how their transcriptomics profiles are analyzed with regard to the hierarchical structure, where the controls provide repeat measurements while the patients provide only one.
  3. It would be informative to describe the number of genes identified after the pre-processing.
  4. Clustering analysis should be better described in the main text. K-means clustering was mentioned in the supplementary material, but it is still not clear how it was applied – PCA followed by K-means clustering only using the 1st and 2nd PCs, I guess?
  5. It would be interesting to report if the DEGs from the two different tissues overlap.
  6. I suggest to rephrase the sentence “… we next sought to answer whether we could also identify diagnostically relevant genes from PBMCs” (lines 262-263), as this study is to explore the transcriptomics profiles across phenotypes, without focus on the diagnostic/predictive power.
  7. Figure S3: it is not clear whether the “size” of the circles means radius or area.

Reviewer 2 Report

The authors describe the results of a very interesting transcriptomic study in adult-onset asthma-related to molds and idiopathic environmental intolerance (IEI). Although sensitization to molds does not seem to be an essential risk factor to develop adult-onset asthma, several similarities have been found between these patients and those with IEI in terms of reported respiratory symptoms and demographic characteristics. Thus, the authors aimed to evaluate the transcriptomic profile of different groups of adult-onset asthma patients and healthy controls in nasal and peripheral blood mononuclear cells. As a result, differences in gene expression levels in both blood and nasal mucosa were detected in asthma patients compared to healthy controls. However, asthma associated with dampness and IEI patients showed similar patterns of gene expression in blood, whereas all asthma regardless of their subclassification and IEI patients semt to be similar in terms of the nasal mucosa transcriptome. Therefore, asthma associated with dampness was revealed to share some similarities with IEI rather with other asthma groups. Evidence of significant enrichment of biological processes in differentially expressed genes in blood was only found in patients with IEI. Moreover, this study suggests the potential role of macrophages in IEI, where the absence of evidence of active airway inflammation has been detected. The authors suggest the potential implication of IEI-related mechanisms in asthma associated with dampness, although further investigation and validation in independent cohorts is needed to confirm this speculation. Furthermore, it is important to appreciate the relevance of the findings of this study suggesting the potential cellular mechanisms of a condition with unknown underlying processes to date such as IEI. This reviewer considers that the authors have correctly conducted the current study and reported their results. I have no further comments.

Author Response

We thank reviewer for the positive overall assessment